# Cutting consumption without diluting the experience: Preferences for different tactics for reducing alcohol consumption among increasing-and-higher-risk drinkers based on drinking context

**Melissa Oldham** [1]*, **Tosan Okpako**[1], **Corinna Leppin**[1], **Claire Garnett**[2], **Larisa-Maria Dina**[1], **Abigail Stevely**[3], **Andrew Jones**[4], **John Holmes**[3]

1 Department of Behavioural Science and Health, University College London, United Kingdom, 2 School of Psychological Science, University of Bristol, United Kingdom, 3 Population Health, School of Medicine and Population Health, University of Sheffield, Sheffield, United Kingdom, 4 School of Psychology, Liverpool John Moores University, Liverpool, United Kingdom

* m.oldham@ucl.ac.uk

## Abstract

Contexts in which people drink vary. Certain drinking contexts may be more amenable to change than others and the effectiveness of alcohol reduction tactics may differ across contexts. This study aimed to explore how helpful context-specific tactics for alcohol reduction were perceived as being amongst increasing-and-higher-risk drinkers. Using the Behaviour Change Technique Taxonomy, context-specific tactics to reduce alcohol consumption were developed by the research team and revised following consultation with experts in behaviour change. In four focus groups (two online, two in-person), N = 20 adult increasing-and-higher-risk drinkers in the UK discussed how helpful tactics developed for four drinking contexts would be: drinking at home alone (19 tactics), drinking at home with partner or family (21 tactics), in the pub with friends (23 tactics), and a meal out of the home (20 tactics). Transcripts were analysed using constant comparison methods. Participants endorsed four broad approaches to reducing alcohol consumption which encompassed all the individual tactics developed by the research team: Diluting and substituting drinks for those containing less alcohol (e.g. switching to soft drinks or no- or low-alcohol drinks); Reducing external pressure to drink (e.g. setting expectations in advance); Creating barriers to drinking (e.g. not buying alcohol to keep at home or storing it in less visible places), and Setting new habits (e.g. breaking old patterns and taking up new hobbies). Three cross-cutting themes influenced how applicable these approaches were to different drinking contexts. These were: Situational pressure, Drinking motives, and Financial motivation. Diluting and substituting drinks which enabled covert reduction and Reducing external pressure to drink were favoured in social drinking contexts. Diluting and substituting drinks which enabled participants to feel that they were having 'a treat' or which facilitated relaxation and Creating barriers to drinking were preferred at home. Interventions to reduce alcohol consumption should

**Data Availability Statement:** The codebook underpinning analysis of the current study is available in the Supplementary Materials.

**Funding:** This study is funded by the Medical Research Council's Public Health intervention Development scheme (MRC grant number MR/W026430/1 to MO). The funders had no role in study design, data collection and analysis, decision to publish, or preparation of the manuscript.

**Competing interests:** JH, AS, CL, TO, LMD declare no conflicts of interest. CG and MO have done paid consultancy work for the behaviour change and lifestyle organization, 'One Year No Beer (OYNB)', providing fact checking for blog posts. OYNB has no links to the alcohol industry or their affiliates. AJ has received funding from CAMARUS pharmaceuticals for unrelated research.

offer tactics tailored to individuals' drinking contexts and which account for context-specific individual and situational pressure to drink.

## Author summary

Reducing alcohol consumption is a public health priority in the UK. The contexts in which people drink are highly variable. This has implications for intervention development as i) Certain drinking contexts may be more amenable to change than others, both in terms of whether people drink at all and how much they drink and ii) Tactics for alcohol reduction could be more or less applicable in different drinking contexts. In this study, increasing-and-higher-risk drinkers discussed alcohol reduction tactics developed by the research team for inclusion in an effective and popular alcohol reduction app, Drink Less. Twenty increasing-and-higher-risk drinkers participated in four focus groups (two online, two in-person). Participants endorsed four broad approaches to alcohol reduction which encompassed the alcohol reduction tactics developed by the research team; Diluting and substituting drinks, Reducing external pressure to drink, Creating barriers to drinking and Setting new habits in the context of an alcohol reduction app. Three cross-cutting themes, Drinking motives, Situational pressure and Financial motivation influenced how applicable these broad approaches, and individual tactics they encompass, were across drinking contexts. This work highlights the importance of accounting for drinking practices and offering tailored support within alcohol reduction interventions.

Alcohol is a dose-dependent [1,2], leading risk factor for preventable cases of cancer and other diseases [3–6] and contributes to health inequalities with the most deprived groups suffering the most harm from alcohol [7]. In the UK, the contexts in which people drink (e.g. socialising in the pub with friends or drinking at home with a partner) are highly variable [8–10]. Some drinking contexts may be more amenable to change than others in terms of whether people drink at all and how much they drink. Furthermore, the applicability of tactics for reducing alcohol consumption may be context dependent. In this study, increasing-and-higher-risk drinkers discussed alcohol reduction tactics developed by the research team and the relative suitability of these tactics in different drinking contexts.

When conceptualising alcohol consumption, researchers have applied theories such as Social Practice Theory, to emphasise the importance of viewing alcohol consumption as an event, occasion or practice-level phenomenon [8,11]. Through this lens what looks like one behaviour, such as drinking a glass of wine, can take on very different 'meanings' in different contexts (e.g. bonding with friends, unwinding after a hard day, or soothing nerves on a first date)[12]. Empirical studies have also identified the need to measure alcohol consumption at an occasion, rather than individual, level. A range of contextual factors are associated with drinking more alcohol within an occasion, including drinking within a large group [13], drinking at the weekend [14] and drinking stronger drinks such as spirits or wine [15]. Other research has identified the predominant types of drinking occasion in Great Britain and Finland (e.g. 'big nights out' and 'drinking at home with family') that account for most alcohol consumption [9,10,16].

Most existing alcohol reduction interventions do not account for variability in drinking practices. Instead, interventions tend to focus on reducing alcohol consumed without

attending to context. However, previous research suggests drinkers do not conceptualise their alcohol consumption in terms of a weekly total, but rather as individual drinking occasions that are differentially integrated, important and acceptable in drinkers' daily lives [17–19]. Tailoring intervention tactics to individuals' drinking contexts, and particularly those contexts in which individuals drink to harmful levels, may be more effective than a 'one-size-fits-all' approach. As such, studies exploring context-specific tactics for alcohol reduction are of value.

Digital interventions, such as software applications ('apps'), offer substantial potential for delivering personalised intervention tactics, while addressing barriers associated with face-to-face interventions and reaching a significant proportion of the population [20]. The Drink Less app is a theory- and evidence-based app [21,22], which resulted in alcohol reduction amongst increasing-and-higher-risk drinkers in a large Randomised Control Trial [23]. For the present study, the research team used the Behaviour Change Technique (BCT) Taxonomy [24,25] to develop context-specific intervention messaging for two of the existing Drink Less components; Insights and Action Planning. The BCT Taxonomy offers a reliable, cross domain, method for specifying, interpreting and implementing the active ingredients of behaviour change interventions (24). For example, within the Action Planning component, the BCT "facilitate goal setting"[24] could be differentially applied to particular drinking contexts. Specifically, someone who consumes most of their alcohol in the pub with friends could be prompted to alternate alcoholic drinks with soft drinks. The two app components were selected as context-specific messaging could be integrated into them straightforwardly and they are regularly used by Drink Less users [26].

We are aware of no research to date which has examined increasing-and-higher-risk drinkers views on the applicability of tactics for alcohol reduction tailored to different drinking contexts. This study used a focus group design to examine these views.

## Design

If focus groups run as intended, a conversational dynamic is established between participants. This facilitates discussion of broad opinions, attitudes, and past experiences [27,28]. This process can lead to participants asking questions and exploring topics and ideas that a researcher in a one-to-one interview may not have broached [28]. Here, the aim of the focus groups was to explore a range of opinions reflecting the experiences of a diverse group of increasing-and-higher-risk drinkers who drink in a range of drinking contexts.

## Materials and methods

The study was designed in line with guidance recommending holding 3–6 focus groups lasting 1–2 hours, each with 6–8 participants and two facilitators [27,28]. This study is reported in line with the Consolidated Criteria for Reporting Qualitative Studies (COREQ) 32-item checklist [29]. The protocol was pre-registered on the Open Science Framework https://osf.io/257t4.

### Ethics

Ethical approval was granted by UCL's Research Ethics Committee (ID: 255627.003). Participants provided informed, written consent prior to participation which was reiterated verbally at the start of focus groups. Identities were removed and data was stored securely.

### Sample

Participants were recruited from an existing database made up of people who have previously taken part in alcohol reduction or smoking cessation studies and given permission for the

research team to recontact them about research studies. Participants were emailed with study information and a link to the screening survey. Participants were also recruited via physical posters around the University campus which featured a QR link to the screening study and digital advertisements on social media accompanied by a link to the screening survey.

Eligibility was determined via screening survey. To be eligible for participation, participants had to be increasing-and-higher-risk drinkers (scoring ≥5 on the AUDIT-C [30]) and interested in using an alcohol reduction app now or in the future. Given the research aims, within each of the focus groups we selected a sample who reported drinking alcohol in a range of different contexts. To ensure the inclusion of diverse viewpoints, the study used a purposive sampling strategy to ensure a maximum variation sample, with representation of different ages and genders, and we aimed to recruit at least half the sample from more disadvantaged socio-economic positions (SEP). This study aimed to recruit six participants for four focus groups (n = 24).

## Setting

Participants were given the option of participating in-person or online. Previous studies have shown that data from online and in-person focus groups is comparable [31] and providing a choice of formats is more inclusive in terms of participants' geographical and socio-economic position. Online participants took part via Microsoft Teams (2 groups) and in-person participants attended on campus at University College London (2 groups). The focus groups were conducted between December 2023-January 2024. MO facilitated discussion, and TO and CL co-facilitated discussion alongside taking observational notes and monitoring recording equipment.

## Development of alcohol reduction tactics and topic guide

To develop context-specific alcohol reduction tactics, it was first necessary to identify the key drinking contexts the tactics should target. A recent typology of drinking occasions identified 15 predominant types of drinking occasion in the UK [32]. The research team simplified this typology to select eight key drinking contexts that could require different tactics to reduce alcohol consumption. These eight contexts and the labels we use to describe them underwent user testing in a previous study and were found to be acceptable and cover most drinking scenarios among increasing-and-higher-risk drinkers [33].

Next, the research team developed context-specific intervention messaging for two existing components of the Drink Less app, i) Insights and ii) Action Planning. This drew on theories (e.g. COM-B [34]) and the Behaviour Change Techniques (BCT) taxonomy [24].

The Insights component gives users weekly feedback on their progress towards meeting their goals. The research team developed messaging which could be delivered within the Insights component. This highlighted the types of contexts individuals were drinking in when they did not meet their goals (e.g. when you drink more than you want to, you tend to be in occasion X).

Within the Action Planning component, users make action plans to facilitate them reaching their goals. These take the form of implementation intentions, or "If. . . Then. . ." plans[35], and can be differentially applied to particular drinking contexts (e.g. someone who consumes most of their alcohol at home alone could be prompted to not buy alcohol to keep at home or buy smaller bottles of alcohol). The team therefore developed suggested action plans (described as alcohol reduction tactics throughout) that the app might prompt the user with, which were specific to eight different drinking contexts[33]: Alone at home, With partner or family at home, Social event in a home, Pub with friends, Pub alone, Big day or night out, Meal

out and Out with Partner. There was some overlap in the tactics between different contexts (e.g. "I will only buy the alcohol I want to drink that day" was relevant for Alone at home and With partner or family at home). These action plans were developed with reference to the BCT taxonomy. For example, the BCT "facilitate goal setting" could be differentially applied to drinking contexts. Someone who consumes alcohol at home alone could be prompted to set goals such as to 'use a measure when pouring spirits or wine' or 'buy smaller package sizes in the supermarket'. Alternatively, an individual more likely to consume multiple drinks in pubs with friends may set goals 'to order soft drinks between alcoholic drinks'. The lead researcher initially developed suggested intervention content for both components, this was then extensively reviewed and edited by the full research team, and wider experts in behaviour change and intervention development in a workshop.

## Procedure

Interested participants consented to the study and completed a screening survey including questions on alcohol consumption (AUDIT-C[30]), willingness to use an alcohol reduction app now or in the future and types of occasions participants typically drank in. Eligible participants were invited to one of four 90-minute focus groups. Consent was reaffirmed at the start of the focus group.

After an icebreaker, there was a short presentation on the Drink Less app, the two relevant components and the plans for the context-specific updates. Participants then discussed how they would feel about receiving feedback on the types of drinking contexts they tended to be in when they did not achieve their goals. Then participants jointly completed a ranking task for two different drinking contexts, putting the alcohol reduction tactics developed by the research team for each context in order of least to most helpful, the aim of this task was to stimulate discussion of each strategy. Throughout the focus groups, the facilitators attempted to ensure that everyone shared views and attempted to draw out differences in opinion by asking whether any participants saw things in a different way. One example of this is following the group ranking task, participants were asked to select the tactics they thought would be personally more or less helpful, this was to draw out differences in opinion within groups. See S1 Appendix for the full topic guide. Each participant was then debriefed and paid a £30 Amazon voucher.

One facilitator took notes during each focus group and afterwards facilitators immediately discussed the topics that arose during the focus groups.

## Analysis

Transcriptions were pseudo-anonymised [gender, age, focus group number], where there were duplicates a, b were added after age. Constant comparison analysis [27,36] of transcripts was then undertaken. Constant comparison enables consecutive analysis of focus groups, to establish whether codes and themes present in earlier groups are seen in later groups.

This involved three stages of coding [36];

1. Open coding–transcripts were read multiple times and codes were attached to chunks of text summarising the topic being discussed.

2. Axial coding–these codes were then grouped into categories with other codes that expressed similar or related topics.

3. Selective coding—themes were developed that expressed the content of the categories.

The analysis approach was a mix of deductive and thematic coding. The intervention content developed by the research team framed much of the discussion in the focus groups and

therefore many of the open codes developed were deductively coded in relation to this. How-ever, participants were also encouraged to discuss tactics they felt were missing or had not been included and new tactics raised by participants were inductively coded. Themes were inductively coded from the data; they were driven by the way participants grouped and talked about different tactics and the factors that were perceived as influencing the applicability of tactics to different settings. When interpreting the categories and formulating themes, the notes taken during focus groups, those documenting conversations directly after focus groups and notes taken during coding were reviewed and this helped inform the structure of the themes. MO undertook preliminary analysis of the first focus group, this was then reviewed by TO and CL with a high level of agreement. There were some suggestions where a code could be applied to new quotes (e.g. mention of self-control that had not been coded as will power). There were two suggestions for where code names should be tweaked to better represent the data (e.g. from "excuses to not drink" to "socially acceptable reasons to abstain"). Finally, there were two suggestions for new codes. One of these, familiarity, representing apparent prefer-ence for tactics participants had previously tried, was adopted. The other suggested code, indi-viduality, was not included as after discussion, we felt this was represented by the coding of dissenting voices for each strategy. Following this, MO reviewed and revised the coding for focus group one, before coding each subsequent focus group in turn. Each stage of this coding was then reviewed and agreed upon by the full research team. Quotes presented in the results section have been edited to remove verbal ticks such as 'umm' and repeated words for clarity. While we recruited fewer participants than originally planned (n = 20 rather than the esti-mated n = 24, though the same number of focus groups was conducted), when analysing data from the last focus group, no new open codes were developed nor did the final focus group change the meaning of any existing codes or themes. As such and in line with previous defini-tions [27], the research team concluded that theoretical and meaning saturation had been achieved. See S2 Appendix for reflections from the researchers on the analysis.

## Results

### Sample characteristics

49 individuals completed the screening survey, 39 were eligible and were emailed to schedule a focus group. 20 participants attended a focus group, with individual focus groups ranging from 4–6 participants due to cancellations on the day. Participant characteristics are shown in Table 1. Focus Groups 2 and 4 took place in-person, whilst Focus Groups 1 and 3 took place online.

**Table 1. Sample Characteristics overall and by focus group.**

| Sociodemographic and Drinking Characteristics | Focus Group 1 (n = 5) | Focus Group 2 (n = 4) | Focus Group 3 (n = 6) | Focus Group 4 (n = 5) | Overall (n = 20) |
|---|---|---|---|---|---|
| Female, % (n) | 60% (3) | 50% (2) | 50% (3) | 80% (4) | 60% (12) |
| Age, m (SD) | 38.8 (12.2) | 53.3 (13.4) | 41.8 (15.1) | 28.0 (3.7) | 39.9 (14.1) |
| Socioeconomic Position, % (n) | | | | | |
| *Live comfortably* | 40% (2) | 50% (2) | 50% (3) | 40% (2) | 45% (9) |
| *Meet needs with a little left* | 40% (2) | 50% (2) | 17% (1) | 60% (3) | 40% (8) |
| *Just meet basic expectations* | 20% (1) | 0% (0) | 33% (2) | 0% (0) | 15% (3) |
| *Don't meet basic expectations* | 0% (0) | 0% (0) | 0% (0) | 0% (0) | 0% (0) |
| AUDIT-C score, m (SD) | 7.4 (1.5) | 9.8 (2.2) | 7.8 (1.8) | 6.2 (0.8) | 7.7 (2.0) |

Notes: m = mean, SD = standard deviation

### Themes

There was a total of 60 codes, which were used to develop 19 categories and seven themes. Four themes focused on broad approaches to alcohol reduction and three themes moderate how applicable reduction approaches are to different contexts. Table 2 presents the themes alongside their categories and codes. A full description of each code can be found in S3 Appendix.

## Overview of themes

Ranking and rating alcohol reduction tactics developed by the research team, resulted in four themes describing broad approaches to reducing alcohol consumption. These were Diluting and substituting drinks, Reducing external pressure to drink, Creating barriers to drinking and Setting new habits. These themes encompassed the alcohol reduction tactics developed by the research team. These four approaches were applied in different ways and were perceived as being differentially helpful across different drinking contexts.

This was in part due to the cross-cutting themes of Situational pressure, Financial motivation and Drinking motives, which differed across drinking contexts. The theme Situational pressure encompassed different forms of pressure to drink, alongside the perceived social costs of reducing alcohol consumption. This theme seemed to be more relevant to social settings, particularly in the context of being in the pub with friends, or situations where a bigger group was present. The Drinking motives theme encompassed different motivations for drinking. In a home context, drinking was often motivated by relaxation. Whereas in larger social contexts drinking was motivated more often by fun or belonging. Drinking for confidence was applied to different settings including pre-drinking at home before a social gathering and in work-related contexts. Finally, Financial motivation impacted on the acceptability of different tactics to reduce consumption across different contexts, being more likely to impact on-trade settings (e.g. bars and pubs).

## Theme 1: Diluting and substituting drinks

Participants discussed different tactics for Diluting and substituting drinks within drinking contexts. This theme encompassed the most tactics; alternating alcoholic drinks with soft drinks or no and low alcohol (no-lo) drinks, drinking lower strength drinks and having smaller drinks or measures.

Alternating alcoholic and soft drinks was perceived as most useful for social occasions in a pub and buying rounds. Whereas having a soft drink or water alongside an alcoholic drink was seen as being more helpful in a home context.

*"That is more phrased for drinking out, where you are out drinking... when you're at home, you could have as many drinks in front of you that you want, of various kinds."* [MALE, 68, FG2]

There were mixed responses to tactics which included switching to soft or no-lo drinks. Some participants highlighted reasons they would avoid soft drinks, including sugar content and the volume of liquid. Other participants highlighted soft drinks they felt were a good replacement for an alcoholic drink, such as kombucha, that felt special and replicated the feeling of a having a treat or a reward at the end of the day.

*"For me, it's like a waste of a beverage having, like, a horrible sickly soft drink that I don't want. Right? So I could have a drink that feels like I'm still experiencing having a beer or glass of wine, but without the consequences."* [FEMALE, 31, FG4]

**Table 2. Themes, categories and codes.**

| Themes (selective coding) | Categories (axial coding) | Codes (open coding) |
|---|---|---|
| *Themes focused on tactics to reduce alcohol consumption* | | |
| 1. Diluting and substituting drinks | Soft drinks | Adult soft drinks<br>Alternate with soft drinks<br>Soft drink at same time<br>Presenting as a drinker |
| | Weaker drinks | Lower strength<br>No and low alcohol (No-Lo)<br>Watering down drinks<br>Drink type |
| | Smaller drinks | Smaller servings<br>Smaller measures |
| | Intentions of dilution/substitution approaches | Drinking fewer drinks<br>Drink slower |
| 2. Reducing external pressure to drink | Reducing expectation to drink | Setting expectations<br>Socially acceptable excuses to abstain<br>Driving rather than drinking<br>Leave early to avoid pressure<br>Avoiding friends who will pressure you |
| | Social support | Social support in sticking to goals<br>Support from partner<br>Concentrating on company |
| 3. Creating barriers to drinking | Limiting availability in the home | Not buying alcohol<br>Only buy what you want that day<br>Out of sight |
| | Reducing opportunity to drink | Making plans in advance<br>Doing something else<br>Time limits<br>Set days for pub |
| 4. Setting new habits | Recognising and breaking old habits | Old habits<br>New information about drinking patterns<br>Will power<br>Pre-drinking<br>Drinking after going out |
| | New routines | Familiarity<br>New habits |
| *Cross-cutting themes impacting on tactics in different drinking contexts* | | |
| 1. Situational pressure | Social cost of reduction | Concerns others will think they have a 'problem'<br>Being perceived as judgemental<br>Problems with friendships<br>Private alcohol reduction goals<br>Not wanting to miss out |
| | Pressure to drink | Pressure to drink<br>Social expectations to drink<br>Social and physical setting<br>People want the fun version of you<br>Rounds |
| | Special occasions | Christmas<br>Birthdays<br>Holidays<br>Weekends |
| 2. Drinking motives | Belonging | Being part of a group |
| | Fun | Drink as a treat<br>Hangovers |
| | Relaxation | Drinking to unwind<br>Ritual of drinking |
| | Confidence | Drinking for confidence |

*(Continued)*

**Table 2.** (Continued)

| Themes (selective coding) | Categories (axial coding) | Codes (open coding) |
|---|---|---|
| 3. Financial motivation | Financial barriers | Value for money<br>Prohibitive cost of no-lo drinks<br>Cheap alcohol at home |
| | Save money by not drinking | Spend money on other things<br>Spending less |

Reponses to drinking lower strength drinks, either through switching from a higher strength drink type such as wine, to a lower strength one such as beer, or by reducing the strength within a beverage category (e.g. from a 6% to a 3% beer or alcohol-free beer), were also mixed. Some felt this would be a helpful strategy, particularly if drinking during the day, whereas others reported not liking the taste of lower strength options.

*"I actually look at what the strength is before I buy a bottle of wine. I don't like the lower strength, that tends to be a bit sweet."* [FEMALE, 60, FG1]

Pouring smaller measures of spirits or wine was highlighted as being particularly helpful in a home context, given some participants felt they overpoured at home. Whereas buying smaller bottles was seen as being more helpful in on-trade contexts, partly due to limited availability of different sized packaging in supermarkets and shops.

*"That doesn't help me because of the type of things that I drink, it's all one size bottles... it would [help] in the pub."* [FEMALE, 29, FG3]

## Cross-cutting themes

The cross-cutting themes impacted on how applicable different dilution and substitution tactics were to different contexts. For example, they were perceived as being helpful for participants in social drinking contexts, in which drinking was more likely to be motivated by "drinking to belong" or "to have fun". Diluting and substituting drinks enabled them to remain part of the social group whilst still limiting their alcohol consumption.

*"I prefer the strategies where you do go to the pub however many times you want and you do go with the friends who like drinking, but you have a strategy to not over drink. [That's] well, I would say would be the ideal, because then you're still having your social life."* [FEMALE, 44, FG2]

Maintaining a presence at social events whilst having a strategy to reduce consumption may also relieve external pressure to drink. Participants did not feel they would miss out and some participants felt that dilution and substitution methods could be done covertly, allowing them to "present as a drinker" helping them avoid being perceived as judgemental and others becoming defensive. However, this was partly dependent on drink and occasion type, with participants feeling it would be easier to pass as a drinker in the pub or in a larger group.

*"Especially if you're in a bigger group.. and if you're drinking things that present as alcohol, you're drinking lower alcohol things, you can probably just sort of like glide through the evening harassment free in some respects, because you're presenting as a drinker."* [FEMALE, 33, FG1]

*"There seems to be a bit of an attitude that if you're somebody who's trying to drink less or you're completely sober, you're very judgmental about people who do drink."* [FEMALE, 33, FG1]

Financial motivations also interacted with dilution and substitution tactics due to a focus on value. This meant buying smaller bottles was often not popular due to the discounts available for larger purchases. Some participants also discussed the prohibitive cost of no-lo drinks, which put them off buying and trying them. Because people attached value to the alcohol content of drinks, they tended to feel that no-los should be cheaper than alcoholic drinks. For some participants, this was exacerbated by previously trying, but not enjoying no-lo drinks. However, many participants did highlight that the range and quality of no-lo drinks had improved in recent years.

*"There are a few, especially if you like sort of craft beers and those sort of hipster beers, there are quite good alternatives now by some big-name brands like BrewDog and that kind of thing. It's just a shame the price doesn't always reflect the fact there isn't any alcohol in them."* [FEMALE, 33, FG1]

## Theme 2: Reducing external pressure to drink

Participants talked about the importance of setting expectations to reduce external pressure to drink alcohol. When drinking with friends this often involved warning people in advance that they would not be drinking or setting a drinking limit. Some participants felt this would be less disappointing to friends and less likely to be perceived as a personal slight or rejection. They also talked about feeling they needed to have socially acceptable reasons for not drinking alcohol, which included working the next day, driving or training for a sports event.

*"I think we've all sort of identified that the social aspect of saying no to a drink can be quite difficult. I think maybe.. like if the app had certain prompts that you could use. I suppose different excuses that maybe would go down better with people? Like I have found just saying no thank you, I don't wanna drink, leads to a lot of questions."* [FEMALE, 33, FG1]

More extreme versions of Reducing external pressure to drink included avoiding certain friends who would pressure them to drink or to leave early if pressured to drink. However, these were less popular options and were seen as a last resort.

### Cross-cutting themes

Situational pressure was particularly relevant to Reducing external pressure to drink. Participants' willingness to ask for social support in their reduction goals often depended on the drinking context or companions. Some participants preferred to ask for support from a partner, rather than friends, although informal ways of doing this were preferred to prevent this from feeling controlling.

*"I think when it's one on one with a close friend, I'd feel a lot more comfortable saying it. But I also... don't think it would be a booze up in the same way if it's one on one, versus if you're going to a big party with a big group"* [FEMALE, 33, FG1]

*"I would be made fun of, whereas with a partner who knows its a serious decision, that'd be fine. I don't have many friends who've raised this with me, so I wouldn't be comfortable raising it with them."* [MALE, 48, FG3]

As this quote indicates, there were some social contexts where participants felt it was less acceptable to say they were not drinking. They felt pressure to be the 'most fun' version of themselves at special occasions and celebrations and felt they would be disappointing friends by being sober, some felt it was 'rude' not to drink at special occasions.

*"It's social situations that are causing me pressure, because I'm usually the life and soul of the party and I'll be coming in with the wine or champagne or whatever, and I am now going to the other side thinking what conversations do I have and how do I go to a wedding. . . saying I'm not drinking?" [FEMALE, 56, FG3]*

These concerns can be understood as 'social costs' of alcohol reduction. Other examples included people feeling that asking for social support would result in negative impacts on friendships or might lead people to think they 'had a problem' with their drinking or were not in control of their drinking. This was often perceived as a severe consequence and something to be avoided. This is in line with the tendency amongst heavier drinkers to construct their drinking identity as positive and healthy, deliberately differentiating themselves from the stigmatised 'alcoholic other'[37].

*"You don't wanna have to say to someone can you help me to control my drinking, because it's something a little bit.. [there's a] weird feeling about that.. is there a problem? Am I not in charge? Am I not in control of that myself?" [MALE, 36B, FG1]*

## Theme 3: Creating barriers to drinking

The third theme focused on Creating barriers to drinking, such as making plans in advance to either limit the availability of alcohol in the moment they might want it or limit whether and how much they drink. This included introducing set start and/or stop times for drinking, or having set days for going to bars or pubs. Others introduced external cues such as pre-booking a taxi or telling people they would leave at a specific time to help them stick to their plans.

*"If I wait for my first drink, that cuts down the number of hours drinking and therefore the number of drinks." [FEMALE, 44, FG2]*

*"I'll usually order a taxi for nine o'clock, so that basically gives me a reason to stop and get back home without getting carried away." [MALE, 61, FG3]*

In home drinking contexts, having less alcohol available in the house, by not buying alcohol, buying only the alcohol that they would drink that day or by storing alcohol in less visible places in the home to avoid temptation, were seen as good barriers to drinking.

*"If you don't have the beers waiting for you in the fridge when you come home, you're less likely to be enticed by them." [FEMALE, 44, FG2]*

### Cross-cutting themes

The cross-cutting theme "Financial motivation" was relevant in whether people created barriers to drinking, with perceived value again playing a role. Only buying alcohol for that day was perceived by some as reducing value for money, as they would not be able to take advantage of multi-pack offers. As above, tactics that were perceived as making alcohol more expensive, particularly in a home context, were generally unpopular.

*"I wouldn't want to commit to not buy multipacks, only buying what I wanted to drink that day or only buying a specific amount because you can save money on bulk purchases and I wouldn't want to stop doing that."* [FEMALE, 44, FG2]

Situational pressure was also relevant. Participants typically reported Creating barriers to drinking in relation to home-drinking or lone-drinking, which were less subject to external social pressure. However, when discussing Creating barriers to drinking in relation to social contexts, participants discussed planning activities which did not centre on alcohol to reduce expectations and pressure to drink. These included going to board game cafes, gyms, and museums. However, participants highlighted that over time alcohol had become more available and had encroached into more activities, such as going to the cinema. This made it harder to identify places where alcohol was not available, and they would experience no pressure to drink. This was also relevant to the Drinking Motives cross-cutting theme as participants talked about finding ways of socialising and having fun with friends without drinking.

*"There's actually so much overlap with alcohol and different settings. . . There used to be the separation of pubs where you went to drink and everywhere else where you went to not drink, or not drink as much. So you would drink with your meal or you would go to the cinema and there wouldn't be any alcohol there."* [FEMALE, 44, FG2]

## Theme 4: Setting new habits

Fewer participants discussed Setting new habits relative to the other themes, and where they did, this tended to be in generalities about the importance of will power in facilitating different tactics. Participants talked about the difficulty of breaking old patterns and routines, the importance of will power in creating new habits and having clear, easy behaviours to implement.

*"[If] I'm stressed I'm gonna pour a glass of wine. That's the point, that's the moment when I probably would need the support."* [FEMALE, 61, FG2]

*"it's OK to say ohh yeah, I will do that. It's the doing.. it's the willpower bit. So it's trying to figure out which is easier, which takes the least willpower or whatever to actually implement."* [MALE, 36B, FG1]

When thinking about how to make new habits stick, participants highlighted the importance of behaviour repetition, having visual prompts to new behaviours and tying new habits to existing behaviours and contexts.

*"If it's somebody who really wants to come in and have another alcoholic drink, if there was no. . . I have got my Horlicks and I'm gonna stick it out on the top.. again, suggestions for what other people have may have done to to break that bit of their habit."* [FEMALE, 61, FG2]

*"setting specific days you know on Mondays I go round mums I won't drink there, easy like just to sort of attach it to another part of your routine."* [FEMALE, 33, FG1]

## Cross-cutting themes

Drinking motives seemed to be related to this theme. Participants spoke about the importance of understanding what was driving their drinking to develop appropriate new habits that

**Table 3. Impact of cross-cutting themes on broader reduction approaches.**

| | Drinking motives | Situational pressure | Financial motivation |
|---|---|---|---|
| Dilution/ substitution | Supports remaining part of the group in upbeat drinking contexts such as night out. Alternatives which feel 'special' or 'different' such as no-lo's or Kombucha can facilitate relaxation in home settings. | Relieves external pressure to drink in larger groups as enables 'covert' reduction and 'passing as a drinker'. | Due to value associated with 'alcohol', no-los and soft drinks perceived by some as poor value for money |
| Reducing external pressure to drink | | Seeking social support can be helpful when with close friends or partner but risks social costs. Social costs are a particular concern in drinking contexts with larger groups and within special occasions where not drinking could be perceived as 'rude', 'judgemental' or disappointing to friends. | |
| Creating barriers to drinking | Planning alternative socialising opportunities to achieve 'fun' and 'belonging' motivations without drinking. | Useful for setting expectations for out of home drinking but increasingly challenging due to reduced number of alcohol-free spaces. | Perceived as poor value for money for restricting home drinking, as missing out on bulk buy or multipack offers. |
| Setting new habits | Understanding motivations can help in new habit formation. | | Shifting money to new habits and hobbies can be a useful facilitator. |

would help them cut down whilst still achieving their desired outcome, whether this was having fun or relaxing.

*"The lower strength alcohol, personally, is useful because I like to unwind on the weekends with a beer. I know I'm gonna want it, but having a lower strength means that it's better for me and I don't feel like I'm missing out." [FEMALE, 33, FG1]*

Financial motivation was also relevant, with some participants discussing redirecting money from alcohol to a new hobby. Though others felt in practice this would be difficult to implement and keep track of.

*"The money I used to spend on alcohol is now in open water swimming and sauna-ing and things like that. So, I made a conscious effort to use my money differently." [FEMALE, 56, FG3]*

See Table 3 for a summary of how the cross-cutting themes impacted on the broader approaches to alcohol reduction.

## Discussion

The research team developed context-specific tactics for alcohol reduction to consider within an alcohol reduction app. Increasing-and-higher-risk drinkers interested in using an alcohol reduction app now or in the future, rated these tactics and endorsed four broad approaches for cutting down. These broad approaches encompassed the alcohol reduction tactics, and were: Diluting and substituting drinks (e.g. through lower strength drinks, smaller drinks or no-lo drinks), Reducing external pressure to drink (e.g. by reducing expectations around drinking or asking for support from friends), Creating barriers to drinking (e.g. by avoiding having alcohol in the house or by setting time limits on drinking) and Setting new habits (e.g. breaking old patterns and taking up new hobbies). Three cross-cutting themes influenced how applicable these approaches were in different types of drinking context; Drinking motives, Situational pressure and Financial motivation.

Understanding the Drinking motives of specific drinking practices can inform tailored tactics which enable behaviour change whilst still facilitating the desired motivation, in healthier

ways[12]. Drinking at home seemed to be more associated with drinking to relax whereas social events, particularly in the on-trade, seemed to be more related to drinking for fun or to belong. As such, in the home approaches that allowed participants to 'have a treat' or to take part in familiar routines, such as no-lo alternatives or adult soft drinks which felt 'special' or 'different' were favoured. For on-trade social events, dilution and substitution tactics, which enabled participants to remain part of the group whilst achieving their reduction goals were favoured. The broad endorsement of Diluting and substituting drinks in different contexts has favourable implications for the role of adult soft drink and no-lo drinks in alcohol harm reduction. No-lo's could potentially be a broadly positive tool to achieve alcohol harm reduction. They may play a similar role to that of vapes in smoking cessation, although there are important distinctions between the two products (e.g. vapes retain the addictive component of smoking while no/lo products remove most or all of the addictive component). However, some expressed concerns about the pricing of no-lo options which served as a barrier for some participants. There are some concerns about no-lo drinks amongst those working in public health, in terms of them sharing marketing and branding with alcohol products[38] and leading to cravings amongst dependent drinkers[39]. Future research examining the role of no-lo's in alcohol harm reduction is required.

The Drinking motives raised in this study draw parallels with an established drinking motives questionnaire[40]. Participants discussed drinking to have fun which mapped on to both 'social motives' and 'enhancement motives'. There was also the drinking motive to belong, which mapped well onto the 'conformity' motivation. Coping motivations were mentioned less frequently. Participants did talk about drinking to relax and for confidence, and three participants mentioned this in the context of feeling stressed, socially anxious or nervous in networking contexts. Previous qualitative research has found that participants tend to blur the line between drinking to relax and drinking to cope [41]. As such, this study may broaden our understanding of how people think and talk about their own drinking. It could be that drinking to relax feels more palatable than drinking to cope, or that coping motivations may be perceived by others as less socially acceptable and might be indicative that they are not in control of their drinking.

Alongside Drinking motives, Situational pressures to drink also differed by drinking context which impacted on the suitability of different reduction approaches. In social, on-trade contexts, dilution and substitution tactics which enabled people to remain part of the 'in-group' of drinkers were preferred to tactics that marked them as a non-drinking other. Particularly with bigger events, participants felt this approach allowed them to engage in covert reduction, where they could 'present as a drinker' and therefore avoid pressure to drink alcohol. Participants differentiated between social expectations to drink, where drinking alcohol was perceived as the default or expected behaviour, and social pressure, more explicit peer pressure from friends. Both contributed to participants feeling that there were social costs to reducing their alcohol consumption within social settings. This was particularly the case in the context of special occasions or celebrations such as birthdays or weddings. This suggests that some drinking contexts, such as special occasions may be less malleable and may require greater levels of intervention than others. Participants experienced less external Situational pressure to drink when drinking at home or alone, where they seemed to be more influenced by habitual patterns of behaviour. This is in line with a previous study, that found that users of an alcohol reduction app, Drink Less, reported they found the app less helpful in controlling their social drinking relative to more habitual home drinking [26]. Some participants liked the idea of teaming up with a partner to help with accountability and reducing temptation. Informal ways of doing this were preferred to prevent this from feeling controlling.

Financial motivation was more likely to facilitate approaches to reduction in on-trade settings and negatively impact tactics for alcohol reduction which reduced value for money in off-trade contexts. Participants felt that they were 'saving money' by drinking in off-trade or home contexts and did not endorse approaches that reduced their value for money. This meant that some tactics falling within the broader approach of Creating barriers to drinking, such as avoiding multipack offers in supermarkets, or dilution or substitution tactics focused on smaller packaging were less favoured by some participants. As mentioned above, the value placed upon the alcohol contained in drinks, also meant that most participants felt that no-lo drinks should be cheaper than their alcoholic counterparts to be considered good value for money. These findings support the need for pricing policy changes which ensure price differentials between no-lo and standard alcohol drinks and remove pricing structures that disincentivise smaller purchases.

An important aspect of this study was that the focus groups took place in December and January. Christmas was described by some participants as being one of the special occasions which presents unique barriers to reduction and many people see January as a time to cut back on drinking through approaches such as Dry January [42]. This could have resulted in participants being more conscious of, or being in the process of trying to reduce drinking, which may have led to a more fruitful discussion. However, it may also be atypical for their usual drinking. We took an inclusive approach by giving participants the option to participate online or in-person. This approach resulted in a geographically varied sample as well as achieving a varied sample in terms of gender and age. However, we had planned to recruit half of participants of a more disadvantaged socioeconomic position. Just under half of our sample reported living comfortably, 40% reported meeting their needs with a little left and 15% were just meeting basic expectations. Unfortunately, we did not recruit participants who were not meeting basic needs. This limits the findings, particularly as one of the themes indicated that financial motivations influenced perceived utility of different approaches and tactics to reduce consumption. As such, those not currently able to meet their needs would likely have had a different perspective. This is something which should be unpicked in future research. It is possible that individuals in this category had less time or availability to take part in research. The more disadvantaged participants that were captured in our sample opted to attend online focus groups, this highlights the importance of providing participants with a choice in future studies. Another limitation of this approach is that the focus groups were made up of increasing-and-higher-risk drinkers who were willing to talk in a group about their alcohol consumption and who were interested in using an alcohol reduction app now or in the future. Therefore, it is likely that these participants are not representative of all increasing-and-higher-risk drinkers both in terms of those experiencing digital exclusion [43], and those not interested in making a future alcohol reduction attempt. Whilst the aim of focus groups is not to be representative it is important to include diverse voices in intervention development research to ensure person-focused and fit-for-purpose interventions, as such alternative methods or recruitment (e.g. by leafleting in less advantaged areas) or data collection (e.g. offering alternative formats or holding focus groups or interviews in more convenient locations) could be explored in future studies. This study centres user voices and highlights broad approaches to alcohol reduction which were deemed as appropriate for different drinking contexts. However, the developed tactics have not undergone efficacy testing, this is now a priority for future research.

## Conclusion

Increasing-and-higher-risk drinkers endorsed four broad approaches to alcohol reduction which encompassed the alcohol reduction tactics developed by the research team; Diluting

and substituting drinks, Reducing external pressure to drink, Creating barriers to drinking and Setting new habits in the context of an alcohol reduction app. Three cross-cutting themes, Drinking motives, Situational pressure and Financial motivation influenced how applicable these broad approaches, and individual tactics they encompass, were across drinking contexts. Dilution and substitution approaches which enabled covert reduction alongside tactics which Reduced external pressure, such as setting expectations in advance, were favoured in social contexts such as in the pub with friends and meals out. Tactics which enabled the broader approach of Creating barriers to drinking, such as limiting alcohol kept in the home and storing alcohol in less visible places, alongside Dilution or substitution tactics such as no-lo alcohol drinks and adult soft drinks which enabled participants to feel that they were having 'a treat' to facilitate relaxation, were preferred in the home.

## Supporting information

**S1 Appendix. Topic Guide.**
(DOCX)

**S2 Appendix. Reflexivity.**
(DOCX)

**S3 Appendix. Qualitative codebook.**
(DOCX)

## Author Contributions

**Conceptualization:** Melissa Oldham, Tosan Okpako, Corinna Leppin, Claire Garnett, Larisa-- Maria Dina, Abigail Stevely, Andrew Jones, John Holmes.

**Data curation:** Melissa Oldham.

**Formal analysis:** Melissa Oldham, Tosan Okpako, Corinna Leppin.

**Funding acquisition:** Melissa Oldham, Claire Garnett, Abigail Stevely, Andrew Jones, John Holmes.

**Investigation:** Melissa Oldham, Tosan Okpako, Corinna Leppin.

**Methodology:** Melissa Oldham, Tosan Okpako, Corinna Leppin, Claire Garnett, Abigail Stevely, Andrew Jones, John Holmes.

**Project administration:** Melissa Oldham.

**Supervision:** Melissa Oldham, John Holmes.

**Validation:** Melissa Oldham, Tosan Okpako, Corinna Leppin.

**Writing – original draft:** Melissa Oldham.

**Writing – review & editing:** Tosan Okpako, Corinna Leppin, Claire Garnett, Larisa-Maria Dina, Abigail Stevely, Andrew Jones, John Holmes.

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
