## [Decision Letter · Decision Letter 0]

4 Jun 2024

PDIG-D-24-00178

Cutting consumption without diluting the experience: preferences for different strategies for reducing alcohol consumption among increasing-and-higher-risk drinkers based on drinking context.

PLOS Digital Health

Dear Dr. Oldham,

Thank you for submitting your manuscript to PLOS Digital Health. After careful consideration, we feel that it has merit but does not fully meet PLOS Digital Health's publication criteria as it currently stands. Therefore, we invite you to submit a revised version of the manuscript that addresses the points raised during the review process.

Please submit your revised manuscript within 60 days Aug 03 2024 11:59PM. If you will need more time than this to complete your revisions, please reply to this message or contact the journal office at digitalhealth@plos.org. Please include the following items when submitting your revised manuscript:

We look forward to receiving your revised manuscript.

Kind regards,

Haleh Ayatollahi

Section Editor

PLOS Digital Health

Journal Requirements:

Additional Editor Comments (if provided):

Reviewers' comments:

Reviewer's Responses to Questions

**Comments to the Author**

1. Does this manuscript meet PLOS Digital Health’s publication criteria? Is the manuscript technically sound, and do the data support the conclusions? The manuscript must describe methodologically and ethically rigorous research with conclusions that are appropriately drawn based on the data presented.

Reviewer #1: Yes

Reviewer #2: Yes

Reviewer #3: Partly

Reviewer #4: Yes

2. Has the statistical analysis been performed appropriately and rigorously?

Reviewer #1: N/A

Reviewer #2: Yes

Reviewer #3: N/A

Reviewer #4: N/A

3. Have the authors made all data underlying the findings in their manuscript fully available (please refer to the Data Availability Statement at the start of the manuscript PDF file)?

Reviewer #1: Yes

Reviewer #2: Yes

Reviewer #3: No

Reviewer #4: Yes

4. Is the manuscript presented in an intelligible fashion and written in standard English?

Reviewer #1: Yes

Reviewer #2: Yes

Reviewer #3: Yes

Reviewer #4: Yes

5. Review Comments to the Author

Reviewer #1: Competent research study of public health and substance use reduction interest, highlighting the importance of accounting for drinking practices and offering tailored support

 within alcohol reduction interventions.

Reviewer #2: I've thoroughly reviewed your paper and offered constructive feedback to enhance its impact. Kindly consider my suggestions to improve clarity and effectiveness. Excited to see the updated version. Best wishes.

Abstract 

1. instead of saying "Some contexts are likely more modifiable than others," consider rephrasing to "Certain drinking contexts may be more amenable to change than others."

2. Provide a brief overview of the methodology used in the study, such as the specific techniques employed in developing context-specific strategies and the rationale behind using focus groups for data collection. This would help readers understand the research approach at a glance.

3. 

Introduction 

1. explicitly state the gap in existing research that this study aims to address. For example, you could emphasize the lack of interventions tailored to individuals' drinking contexts and the potential effectiveness of such tailored approaches.

2. When discussing the variability of drinking contexts and the importance of viewing alcohol consumption as a social practice, consider providing specific examples to illustrate these points. For instance, you could describe typical drinking occasions in different contexts, such as social gatherings with friends or relaxing evenings at home.

3. why such interventions are particularly relevant and effective for addressing alcohol reduction. Highlighting the accessibility, scalability, and potential for personalization of digital interventions would strengthen this argument.

4. Provide a brief explanation of why the Behaviour Change Technique (BCT) Taxonomy was chosen as the framework for developing context-specific intervention messaging. Discuss how this taxonomy helps identify effective behavior change techniques and tailor interventions to specific contexts.

5. Where relevant, connect the current study to previous research on alcohol reduction interventions, digital health interventions, and the effectiveness of context-specific approaches. This would help situate the study within the existing literature and highlight its novelty and contribution to the field.

6. Clearly articulate what makes this study novel and innovative compared to previous research. This could include its focus on context-specific interventions, the use of the BCT Taxonomy for intervention development, or its exploration of increasing-and-higher-risk drinkers' perspectives on alcohol reduction strategies.

Method

1. Provide more details about the purposive sampling strategy used to select participants. Explain how participants were recruited from the existing database and the rationale behind selecting individuals with different ages, genders, and socioeconomic positions. 

2. Expand on the rationale for choosing focus groups as the primary method for data collection. Discuss why focus groups were considered suitable for exploring participants' perspectives on alcohol reduction strategies in different contexts, particularly among increasing-and-higher-risk drinkers. 

3. Offer additional details about the data collection process, including how focus groups were conducted, how facilitators managed group dynamics, and any strategies used to ensure participants felt comfortable sharing their opinions. 

4. Provide a clearer explanation of how context-specific intervention messaging was developed for the Drink Less app. Discuss the theoretical frameworks and behavior change techniques used to inform the development of these messages and how they were tailored to different drinking contexts. 

5. Provide a more detailed explanation of the data analysis approach, particularly the constant comparison analysis method. Describe how the coding process was conducted, including the steps involved in open coding, axial coding, and selective coding. 

6. explain how themes and broader approaches were derived from the data and how the analysis incorporated both deductive and inductive coding approaches.

7. Provide a clearer discussion of data saturation and how it was assessed in the study. Explain how theoretical saturation was reached, particularly given the smaller sample size than originally planned. 

Results

• Provide more context about the screening survey and eligibility criteria to better understand the participant selection process.

Discussion 

• While the four broad approaches are mentioned, provide concise definitions or examples for each approach. This ensures readers have a clear understanding of what each strategy entails. For instance, expand on what "Setting new habits" entails—does it involve replacing drinking routines with healthier alternatives, or incorporating non-drinking activities into social events?

• While the text references previous research on drinking motives and situational pressures, integrate these findings more seamlessly into the discussion.

• Rather than simply mentioning parallels with established questionnaires, discuss how these findings contribute to a deeper understanding of participants' behaviors and preferences.

• offer potential solutions or avenues for future research to address these concerns, such as exploring alternative pricing strategies for no-lo drinks or implementing targeted recruitment methods to include diverse participant demographics.

• provide recommendations for app developers, policymakers, or public health practitioners based on the study's insights would enhance the conclusion's relevance and applicability.

Reviewer #3: This study on the context-dependency of strategies for reducing alcohol consumption is highly interesting and much needed. There are some issues with the manuscript that need to be addressed before this work can be considered publishable, though. The qualitative perspectives should have more voice in the study of such a complex phenomenon as alcohol consumption. I also have a major concern with the conceptional framing of the manuscript, as well as some minor suggestions for improvements.

The major issue is the application of the concept “strategies” in the context of social practices. Here, I can see some theoretical misalignments. Even in the table of thematic coding, the strategies are quite central. Is it appropriate to call it strategies when the final act of the social practice of alcohol consumption should be on the individual level? Would that not rather be a tactic? In a nutshell, strategies are top-down and initiated by those governing populations to governmentalize their behaviour while tactics are bottom-up and employed situationally by individuals to navigate social situations. As an example of this, consider how this distinction is employed in work on patient empowerment:

https://doi.org/10.1177/1363459319831343

This is not to say that this manuscript needs to follow the same path, but it would be advisable to clearly position the use of the concept “strategies” (such as in “coping strategies” a la Lazarus, R. S., & Folkman, S. (1984). Stress, appraisal, and coping. Springer publishing company) or employ another term such as “tactics”.

Detailed suggestions for improvement and strengthening the contributions of the manuscript:

• Please, reformulate line 88-89: Theories, including Social Practice Theory, highlight the importance of viewing alcohol consumption as an event, occasion or practice-level phenomenon (8,11). Both references are applying the theory and not necessarily presenting the one.

• Lines 89-91 please, reformulate or provide references to this statement : “This requires researchers to move away from thinking about alcohol consumption as a single health behaviour and to instead think of alcohol consumption as one constituent part in a range of social practices.” Which researchers say this? How can alcohol consumption be a health behavior when research is every day more strongly agreeing on the negative effects and consequences of such? Maybe “a health- related behaviour”? Or “health-related practices with direct impact on the health outcomes”?

• More reflections on the different contextualization of the methods applied are needed. The F2F focus groups are not the same as the technologically mediated ones. What limitations and differences are there?

• Regarding the coding, can the three-stage approach be explained in some more detail? Please also add a discussion of how the themes raised during the focus groups that the facilitator uncovered were employed in the coding process?

• Females are overrepresented in both the sample and the data presentation. Please reflect on the reasons and implications. Also, how as “female”established? Self-reported or observed? How does it impact the results of the study? Such reflections should be integrated into limitations or discussion sections.

Reviewer #4: Peer Review for: Cutting consumption without diluting the experience: preferences for different strategies for reducing alcohol consumption among increasing-and-higher-risk drinkers based on drinking context.

General Comments

I commend the authors for their valuable contribution to the field of alcohol intervention research. The use of focus groups to identify mechanisms that could aid in reducing alcohol consumption and integrating patient perspectives into the development of an intervention app is highly commendable and enriches the field. Including the patient voice in research not only increases the study's relevance but also enhances the potential for real-world applicability. Generally speaking, too little qualitative research that focuses on including the lived-experience of patients is still being conducted, and this article takes a good step into this direction. 

 Strengths

Patient-Centered Approach: The inclusion of patient voices through focus groups is a significant strength. This approach ensures that the strategies identified are grounded in real-world experiences and needs, which is crucial for developing effective interventions.

Development Process: The methodical process of using qualitative data to inform the development of an intervention app is well-documented and provides a clear roadmap for future research in this area.

Limitations

Lack of Efficacy Testing: A major limitation of the study is that the suggested strategies have not been tested for efficacy. This should be explicitly stated as it limits the research's usefulness for other researchers who may wish to build on this work. Without efficacy data, it is challenging to assess the potential impact of these strategies on reducing alcohol consumption.

Variation in AUDIT Scores: The focus groups appeared to differ in their Alcohol Use Disorders Identification Test (AUDIT) scores. It would be beneficial to conduct significance testing to determine whether these differences are statistically significant. If significant differences are found, presenting the outlier group separately could provide valuable insights into whether participants with particularly high AUDIT scores offer different suggestions.

Specific Recommendations

Explicit Limitation on Efficacy: Clearly state the limitation regarding the lack of efficacy testing in the discussion section. This transparency will help other researchers understand the scope and applicability of your findings.

Significance Testing for AUDIT Scores: Perform statistical analyses to determine if the focus groups differ significantly in their AUDIT scores. If significant differences are present, consider discussing the implications of these differences and presenting the data from the outlier group separately. This additional analysis could uncover nuanced insights and enhance the depth of your findings.

 Overall, I recommend this article for publication, subject to the authors addressing the noted limitations and considering the suggested analyses. The study makes a valuable contribution to the field, and with these enhancements, it can offer even greater utility to researchers and practitioners working on alcohol intervention strategies.

6. PLOS authors have the option to publish the peer review history of their article (what does this mean?). If published, this will include your full peer review and any attached files.

**Do you want your identity to be public for this peer review?** For information about this choice, including consent withdrawal, please see our Privacy Policy.

Reviewer #1: No

Reviewer #2: Yes: Roghieh Nooripour

Reviewer #3: No

Reviewer #4: No

---

## [Decision Letter · Decision Letter 1]

10 Jul 2024

Cutting consumption without diluting the experience: preferences for different strategies for reducing alcohol consumption among increasing-and-higher-risk drinkers based on drinking context.

PDIG-D-24-00178R1

Dear Dr Oldham,

We are pleased to inform you that your manuscript 'Cutting consumption without diluting the experience: preferences for different strategies for reducing alcohol consumption among increasing-and-higher-risk drinkers based on drinking context.' has been provisionally accepted for publication in PLOS Digital Health.

Best regards,

Haleh Ayatollahi

Section Editor

PLOS Digital Health

Reviewer Comments (if any, and for reference):

Reviewer's Responses to Questions

**Comments to the Author**

1. If the authors have adequately addressed your comments raised in a previous round of review and you feel that this manuscript is now acceptable for publication, you may indicate that here to bypass the “Comments to the Author” section, enter your conflict of interest statement in the “Confidential to Editor” section, and submit your "Accept" recommendation.

Reviewer #1: All comments have been addressed

Reviewer #2: All comments have been addressed

Reviewer #3: All comments have been addressed

Reviewer #4: All comments have been addressed

2. Does this manuscript meet PLOS Digital Health’s publication criteria? Is the manuscript technically sound, and do the data support the conclusions? The manuscript must describe methodologically and ethically rigorous research with conclusions that are appropriately drawn based on the data presented.

Reviewer #1: Yes

Reviewer #2: Yes

Reviewer #3: Yes

Reviewer #4: Yes

3. Has the statistical analysis been performed appropriately and rigorously?

Reviewer #1: Yes

Reviewer #2: Yes

Reviewer #3: N/A

Reviewer #4: Yes

4. Have the authors made all data underlying the findings in their manuscript fully available (please refer to the Data Availability Statement at the start of the manuscript PDF file)?

Reviewer #1: Yes

Reviewer #2: (No Response)

Reviewer #3: No

Reviewer #4: Yes

5. Is the manuscript presented in an intelligible fashion and written in standard English?

Reviewer #1: Yes

Reviewer #2: Yes

Reviewer #3: Yes

Reviewer #4: Yes

6. Review Comments to the Author

Reviewer #1: I am happy with the revisions made.

Reviewer #2: After reviewing the changes made by the authors, I can see that they have attended to the revisions appropriately and the manuscript seems acceptable to be published within this form.

Reviewer #3: The manuscript has been revised thoroughly, addressing all the concerns I raised during the first round of reviews.

Reviewer #4: (No Response)

7. PLOS authors have the option to publish the peer review history of their article (what does this mean?). If published, this will include your full peer review and any attached files.

**Do you want your identity to be public for this peer review?** For information about this choice, including consent withdrawal, please see our Privacy Policy.

Reviewer #1: No

Reviewer #2: **Yes: **Roghieh Nooripour

Reviewer #3: No

Reviewer #4: No
